# Research on a Novel Hybrid Decomposition–Ensemble Learning Paradigm Based on VMD and IWOA for PM_2.5_ Forecasting

**DOI:** 10.3390/ijerph18031024

**Published:** 2021-01-24

**Authors:** Hengliang Guo, Yanling Guo, Wenyu Zhang, Xiaohui He, Zongxi Qu

**Affiliations:** 1School of Geoscience and Technology, Zhengzhou University, Zhengzhou 450001, China; guohengliang@zzu.edu.cn (H.G.); yuzhang@lzu.edu.cn (W.Z.); 2College of Atmospheric Sciences, Lanzhou University, Lanzhou 730000, China; guoyl17@lzu.edu.cn; 3School of Management, Lanzhou University, Lanzhou 730000, China; quzx@lzu.edu.cn

**Keywords:** PM_2.5_ prediction, ensemble model, weight coefficient optimization, whale optimization algorithm

## Abstract

The non-stationarity, nonlinearity and complexity of the PM_2.5_ series have caused difficulties in PM_2.5_ prediction. To improve prediction accuracy, many forecasting methods have been developed. However, these methods usually do not consider the importance of data preprocessing and have limitations only using a single forecasting model. Therefore, this paper proposed a new hybrid decomposition–ensemble learning paradigm based on variation mode decomposition (VMD) and improved whale-optimization algorithm (IWOA) to address complex nonlinear environmental data. First, the VMD is employed to decompose the PM_2.5_ sequences into a set of variational modes (VMs) with different frequencies. Then, an ensemble method based on four individual forecasting approaches is applied to forecast all the VMs. With regard to ensemble weight coefficients, the IWOA is applied to optimize the weight coefficients, and the final forecasting results were obtained by reconstructing the refined sequences. To verify and validate the proposed learning paradigm, four daily PM_2.5_ datasets collected from the Jing-Jin-Ji area of China are chosen as the test cases to conduct the empirical research. The experimental results indicated that the proposed learning paradigm has the best results in all cases and metrics.

## 1. Introduction

Pollution of the environment is one of the most serious issues facing humankind today, and badly polluted air can cause great damage in economics and people’s lives. According to the World Health Organization (WHO), it is known that almost 3 million children die every year from a range of problems caused by air pollution [1]. With the process of industrialization and urbanization, the air pollution is becoming increasingly serious and the hazy weather has grown rapidly, especially in developing countries. In recent years, the foggy weather in many areas of China have become increasingly serious. Since the beginning of 2013, sustained haze weather has turned Beijing-Tianjin-Hebei (Jing-Jin-Ji region) into heavy pollution region. Fine particulate matter is one of the key contributors that leading to air pollution and hazy weather. It carries many adverse health effects, such as respiratory diseases and premature death [2].

Recently, increasingly countries have set up environmental monitoring systems, which can provide a large amount of PM monitoring data. However, PM data are affected by many factors and fluctuates greatly over time, making it very challenging to predict. Therefore, many models and tools have been developed to predict PM_2.5_ and other air pollutant concentrations to improve the accuracy of the predictions. These models can be generally categorized into physical, statistical and hybrid models. For example, physical methods can be used to simulate the processes of emissions, diffusion and transfer of pollutants through meteorological, emission, and chemical models [3,4,5]. Statistical methods which mainly include autoregressive integrated moving average model (ARIMA), artificial neural networks (ANN) and multiple linear regression (MLR) [2,6,7,8,9], have been broadly applied to the pollutant concentration prediction. For instance, Ref. [10] proposed a forecasting model based on MLR and bivariate correlation analysis to predict the annual and seasonal concentrations of PM10 and PM_2.5_. Ref. [11] studied the effects of meteorological factors on ultrafine particulate matter (UFP) and PM10 concentrations under traffic congestion conditions using the ARIMA model. However, in practice, most pollutant sequences are non-linear and irregular, which may involve the problem of non-linear dynamical systems, so these linear algorithms are still problematic in predicting PM concentration. On the contrary, using artificial neural network models to predict pollutant concentration can overcome the limitations of traditional linear models and handle nonlinear problems well. [12] developed extended model based on long-term and short-term memory neural network. The model takes into account the spatiotemporal correlation to predict the pollutant concentration and shows excellent performance. [13] applied cuckoo search (CS) to optimize BPNN to predict PM concentrations in four major cities in China.

Recently, to predict air quality more accurately, many hybrid models have been proposed based on ensemble learning paradigms, data preprocessing techniques and heuristic algorithms. For example, Ref. [14] developed a new prediction model based on the multidimensional k-nearest neighbor model and the ensemble empirical mode decomposition (EEMD) method. Ref. [15] developed a novel hybrid model based on wavelet transform (WT) and stacked autoencoder (SAE) and long short-term memory (LSTM) to simulate PM_2.5_ at six sites in China. Ref. [16] developed a model based on a combination of WT and neural network algorithm to decompose the PM_2.5_ data and then perform sub-series prediction analysis and finally data reconstruction. Ref. [17] proposed a novel PM_2.5_ hybrid prediction model, which includes a new pre-processing method (wavelet transform and variational mode decomposition), using differential evolution (DE) algorithm optimized BPNN to predict each decomposition sequence. The drawback of the decomposition-based prediction model is that using a single method to predict all signal sequences. Since different decomposition sequences have different characteristics, a single model does not fit all the characteristics of the decomposition sequences [18]. Thus, ensemble prediction model integrated multiple single models will help avoid the shortcomings of a single model and further improve the prediction accuracy. Furthermore, many heuristic algorithms are used to help optimize the weight coefficients of the ensemble model. [19] developed an ensemble model based on differential evolution (DE) to determine the optimum weights for electricity demand forecasting. Ref. [20] employed the cuckoo search algorithm (CSO) to optimize the weight coefficients of ensemble model. Whale optimization algorithm (WOA), proposed by Ref. [21], is a novel heuristic algorithm by imitating whale behavior in nature. However, the WOA will encounter problems such as being stuck in a local optimal solution and slow in convergence, when solving more complex problems. Thus, a new improved whale optimization algorithm (IWOA) is proposed in this study to strengthen the local seeking capability of the WOA.

Through the above analysis, considering the criticality of data pre-processing and the limitations of one single prediction model, a new hybrid decomposition–ensemble learning paradigm based on variation mode decomposition (VMD) and modified whale-optimization algorithm (IWOA) is introduced. First, the original PM sequence is decomposed into different VM sequences using VMD. Then, the weight-determined ensemble model, which optimized by IWOA, is employed to forecast each decomposition component. Finally, several prediction subsets are assembled into the final prediction result.

The paper is structured as follows: in Section 2, several single forecasting models, the ensemble prediction theory and VMD, are introduced. In Section 3, the proposed decomposition–ensemble model is presented. In Section 4, the study areas and the evaluation criteria are described. In Section 5, the comparative results of the proposed model and other models is in conducted. Finally, in Section 6, the conclusions the important results of this paper are explicitly introduced.

## 2. Related Methodology

Four individual forecasting models, VMD, IWOA, which employed in the suggested ensemble model, are described as follows.

### 2.1. Four Individual Prediction Methods

In latest years, many prediction models have been developed and applied to PM_2.5_ concentration prediction. This paper uses four popular methods, BPNN, ANFIS, ANFIS-FCM and GMDH, which show good performance in PM_2.5_ prediction, to construct the ensemble models.

#### 2.1.1. The Back Propagation Neural Network (BPNN)

The BPNN is a multi-layer feed-forward neural network, which is widely used in many fields. The BPNN algorithm needs to find the parameter with the minimum error, i.e., the minimum value of the error between the output value and the actual value according to the negative gradient direction. The process of the BPNN is mainly divided into update and learning stages:(1)$$k_{ij}\left(t\right)=w_{ij}\left(t-1\right)-\Delta w_{ij}\left(t\right)$$
(2)$$\Delta w_{ij}\left(t\right)=\eta \partial E/\partial w_{ij}\left(t-1\right)+\alpha \cdot \Delta w_{ij}\left(t-1\right)$$
where *η* denotes the learning speed, *w_ij_* represents the weights between nodes *i* and *j*, *α* denotes the impulse parameter, *E* denotes the error super curve face and *t* denotes the current iterative steps.

#### 2.1.2. The Adaptive Network Based Fuzzy Inference System (ANFIS)

Ref. [22] proposed the ANFIS that combines the blur systems and neural networks. It plays the advantages of both and makes up for the shortcomings of each. ANFIS can form an adaptive neuro-fuzzy controller by using a neural network learning mechanism to automatically retrieve rules from the input and output sample data. Through the offline training and the online learning algorithms, it can create fuzzy inferences and control the self-adjustment of rules, thereby making the system itself develop towards adaptive, self-organizing, and self-learning. ANFIS includes five-layer network, and each layer contains several node functions.

Layer 1: This process is the fuzzy layer. Each node in layer 1 is adaptive, all have node function, and will generate membership of a fuzzy set.
(3)$$O_{i}^{1}=\mu_{A_{i}}\left(x\right)$$
where *x* denotes the input to node *I*, and *A_i_* denotes the language label associated with the function of this node. The “*μ*” denotes the membership functions for *A_i_*, which described by generalized Gaussian functions.

Layer 2: In this process every node is a circular node labeled ∏ out, i.e., ∏-norm operation:(4)$$O_{i}^{2}=\mu_{A_{i}}\left(x\right)+\mu_{B_{i}}\left(y\right),i=1,2$$

Layer 3: At each node in this layer, the ratio of the firing weights under the *i*th rule to the sum of the firing weights under all rules is calculated:(5)$$O_{i}^{3}={\bar{w}}_{i}=\frac{w_{i}}{w_{1}-w_{2}},i=1,2$$

At this level, all outputs are collectively referred to as normalized emission intensity.

Layer 4: In this process, the contribution of the *i*th rule to the overall output is calculated:(6)$$O_{i}^{4}={\bar{w}}_{i}f_{i}={\bar{w}}_{i}\left(a_{i}x+b_{i}y+c_{i}\right),i=1,2$$
where ${\bar{w}}_{i}$ represents the out of layer 3, and $\left(a_{i}x+b_{i}y+c_{i}\right)$ is the parameter set.

Layer 5: In this layer, the signal node ∑ calculate the final output as the sum of all incoming signals:(7)$$O_{i}^{5}=\sum_{i}{\bar{w}}_{i}f_{i}=\frac{\sum_{i}w_{i}f_{i}}{\sum_{i}w_{i}}$$

The final output of the adaptive neural fuzzy inference system is:(8)$$\begin{array}{l} f_{out}={\bar{w}}_{1}f_{1}+{\bar{w}}_{2}f_{2}=\frac{w_{1}}{w_{1}+w_{2}}f_{1}+\frac{w_{2}}{w_{1}+w_{2}}f_{2}\\ =\left({\bar{w}}_{1}x\right)p_{1}+\left({\bar{w}}_{1}x\right)q_{1}+\left({\bar{w}}_{1}x\right)r_{1}+\left({\bar{w}}_{2}x\right)p_{2}+\left({\bar{w}}_{2}x\right)q_{2}+\left({\bar{w}}_{2}x\right)r_{2} \end{array}$$

#### 2.1.3. The Fuzzy C-Means Clustering (FCM)

The FCM is a type of data aggregation method. In the FCM method, each data point needs to be classified as a level assigned at the member level in the cluster. FCM divides a selection of n vector *x_i_*, (*i* = 1, 2, …, *n)* into fuzzy groups, and then finds a clustering center in each fuzzy group in a way that minimizes the cost function of the similarity measure. The above *i* = 1, 2, …, *c* represents random sampling from n points. Here is a brief introduction to the stage of the FCM algorithm. First, choosing the centers of cluster *c_i_*, (*i* = 1, 2, …, *c*) from the *n* points (*x*_1_, *x*_2_, *x*_3_, *…, x_n_*) randomly. Second, the membership matrix *U*, is calculated by using the subsequent equation as follows:(9)$$\mu_{ij}=\frac{1}{\sum_{k=1}^{c}{\left(\frac{d_{ij}}{d_{kj}}\right)}^{\frac{2}{m-1}}}$$
where $d_{ij}=\left\|c_{i}-x_{j}\right\|$ is the Euclidean distance which involves the *i-th* cluster center and the *j-th* data point, and *m* is the fuzziness index. Third, compute the cost function using the following formula.
(10)$$J\left(U,c_{1},\ldots ,c_{2}\right)=\sum_{i=1}^{c}j_{i}=\sum_{i=1}^{c}\sum_{j=1}^{n}\mu_{ij}^{m}d_{ij}^{2}$$

Stop the process when it falls below a certain threshold. Additionally, finally, the new *c* fuzzy clustering center *c_i_*, *i* = 1, 2, …, *c* is calculated using the following equation:(11)$$C_{i}=\frac{{\sum_{j=1}^{n}\mu }_{ij}^{m}x_{j}}{{\sum_{j=1}^{n}\mu }_{ij}^{m}}$$

#### 2.1.4. The Group Method of Data Handling (GMDH)

The GMDH is a series of computer-based inductive algorithms for the mathematical modeling of multi-parameter data sets. It is characterized by a fully automatic structure and parameter optimization of the model. The GMDH can be used for data extraction, knowledge detection, prediction, modeling of complex systems, optimization, and pattern recognition [23]. The GMDH algorithm features an induction procedure to classify increasingly complex multinomial models and select the best solution by the externality criterion.

The GMDH pattern generally have multiple sets of inputs and one set of outputs, and is a subset of the components of the basic function:(12)$$Y\left(x_{1},\ldots ,x_{n}\right)=a_{0}+\sum_{i=1}^{m}a_{i}f_{i}$$
where *a* denotes coefficients and *f* denotes the fundamental function that depends on different inputs, *m* represents the number of fundamental function components.

The basic function (12) is called the partial model and the GMDH considers various subsets of this function and thus finds the optimal solution. The coefficients of the model are first estimated using the least squares method. Then, the number of local components of the model is gradually increased. Finally, the GMDH algorithm finds the best complexity model structure by minimizing the external criterion. This process is called self-organizing of the model.

The main idea of the GMDH neural network learning algorithm is as follows: a series of source neurons are generated by performing cross-combining on each entry unit of the system, and the mean square error of the output error corresponding to each neuron is calculated; then, several outputs are chosen from the created neurons with smaller mean square error than a predetermined threshold, the selected neurons are used as the input unit of the new generation; the process of survival of the fittest and gradual evolution is repeated until the new generation of neurons is no better than the previous generation.

### 2.2. Variation Mode Decomposition (VMD)

The VMD is a non-recursive signal treatment algorithm that decomposes the original signal into a family of patterns with a specific frequency spectrum domain bandwidth [24]. During the decomposition process, each pattern can be compressively pulsed around a certain center. If the bandwidth of each pattern is required, three steps should be completed. After that, a constraint variational problem can be given. The details of VMD are described in [25].

### 2.3. Optimization Algorithm-IWOA

An improved heuristic algorithm IWOA is developed to enhance the performance of the ensemble model. The IWOA determines the optimal weight coefficient of the ensemble model. The basic whale optimization algorithm (WOA), chaotic local search (CLS), and the WOA modified by CLS will be described below.

#### 2.3.1. Overview of the Whale Optimization Algorithm

The WOA, put forward by S. Mirjalili in 2016, is a simulation of the hunting mechanism of humpback whales, called bubble-net feeding method. Humpback whales form distinctive bubbles by circling around their prey in a circular or “9 shaped” path during foraging. With special exercises, humpback whales first form a spiral bubble 10-15 m below their prey while swimming upstream to the surface. Then, it surrounds the prey with its flashing fins to prevent it from escaping and catch it [21]. The mathematical principles of the above humpback whale behavior are described as follows:

(a) Encircling prey:

The humpback whale orbits its prey, and updates its position to the best search agent as the number of iterations increases. It can be depicted mathematically as:(13)$$\vec{D}=\left|\vec{C}\vec{X^{*}}(t)-\vec{X}(t)\right|$$
(14)$$\vec{X}(t+1)=\vec{X^{*}}(t)-\vec{A}\cdot \vec{D}$$
where *X** denotes the position vector of the best solution obtained thus far, $\vec{X}$ is the position vector, $\vec{A}$ and $\vec{C}$ denote the coefficient vectors, and *t* denotes the current iteration.

(b) Bubble-net attacking method:

The mathematical modeling of humpback whale’s vesicular behavior is designed for the following two methods. 1. Shrinking encircling mechanism: This is a bracket predation mechanism that requires finding a new agent location, which can be anywhere between the agent’s original location and the current optimal agent location. The values of $\vec{A}$ in this process is in the interval [−1, 1].

2. Spiral update position: A spiral equation needs to be created between the positions of both the prey and the whale to mimic the spiral motion of the humpback whale, as shown below.
(15)$$\vec{X}\left(t+1\right)=\vec{D^{\prime }}\cdot e^{bl}\cdot \cos\left(2\pi l\right)+\vec{X^{*}}\left(t\right)$$

The probability *p*, a random number in [0,1], is assumed to select between the shrinking encircling and the spiral-shaped path during the optimization process.

(c) Search for prey:

During the search phase, the variation of vector $\vec{A}$ can also be used to find prey at random. Therefore, to move away from a reference whale, $\vec{A}$ can be utilized with the random values greater than 1 or less than −1. The mathematical model at this stage is as follows:(16)$$\begin{array}{l} \vec{X}\left(t+1\right)=\vec{X_{rand}}-\vec{A}\cdot \vec{D}\\ \vec{D}=\left|\vec{C}\cdot \vec{X_{rand}}-\vec{X}\right| \end{array}$$
where $\vec{X_{rand}}$ is a random position vector (random whale) selected from the current population.

#### 2.3.2. IWOA

As mentioned earlier, WOA was recently proposed and widely used in many fields. However, it also has shortcomings, like slow conversion in the late stage and easy to fall into a local optimum. Additionally, the chaotic local search (CLS), based on chaotic search, can effectively avoid the local optimization and converge to the global optimization. The blending of WOA and CLS can help improve global conversion and prevent falling into local solutions. To accelerate the local convergence of WOA, the chaotic local search algorithm is also applied. When WOA finishes iterating to find the best solution, the acceptance of these new solutions as determined by CLS will perform to local search a better solution close to the best solution. A logistic equation applied in CLS is defined as follows:(17)$$cx_{i}^{iter+1}=\mu cx_{i}^{iter}\left(1-cx_{i}^{iter}\right)$$
where $cx_{i}$ denotes the *i*th chaotic variable, *iter* is the iteration number. When μ=4, the above equation exhibits chaotic dynamics, $cx_{i}$ denotes range in (0,1) and $cx_{0}\notin \left\{0.25,\;0.5,\;0.75\right\}$. For more details about CLS, please refer to [26].

The pseudo-code of the IWOA algorithm is outlined as follows:
**Algorithm: Improved whale-optimization algorithm (IWOA)***Objective:*Minimize and maximize the objective function f(x), $x_{i}=\left(x_{i1},\;x_{i2},\;\ldots ,\;x_{id}\right)$*Parameters:**iter*-iteration number.*Maxiter*-the maximum number of iteration.*I*-a population pop.*p*-the switch probability**1.** /*Initialize a population $x_{i}=\left(x_{i1},\;x_{i2},\;\ldots ,\;x_{id}\right)$**2. WHILE***iter* < *Maxiter***3.** **FOR***i* = 1 to *I* *Update*
$\vec{A}$, $\vec{C}$, *l* and *p***4.** **IF***p* > 0.5**5.** **IF**$\left(\left|\vec{A}\right|<+1\right)$**6.** Update the position of the current solution by Equation (14)**7.** **ELSE IF**$\left(\left|\vec{A}\right|>+1\right)$**8.** Randomly choose a search agent**9.** Update the position of the current search agent by Equation (16)**10.** **END IF****11.** **ELSE IF***p* > 0.5**12.** Update the position of the current search by Equation (15)**13.** **END IF****14.** **END FOR****15.** /*Jump out of local optimum by using chaotic local search. */**16.** Calculate $cx_{i}^{iter}=\frac{x_{i}^{iter}-x_{i}^{\min}}{x_{i}^{\max}-x_{i}^{\min}}$**17.** Calculate the next iteration chaotic variable by Equation (16)**18.** Transform $cx_{i}^{iter+1}$ for the next iteration $x_{i}^{iter+1}=x_{i}^{\min}+cx_{i}^{iter+1}\left(x_{i}^{\max}-x_{i}^{\min}\right)$**19.** /*Evaluate $x_{i}^{iter}$ replace $x_{i}^{iter}$ by $x_{i}^{iter+1}$ if the newly generation is better. */**20.** /*Find the current best solution *gbest**/**21.** iter = iter + 1**22.** **END WHILE**

## 3. Decomposition–Ensemble Learning Paradigm

In this part, we suggest a new hybrid decomposition–ensemble learning paradigm that integrates VMD method, several prediction models and IWOA optimization. The main process of the developed decomposition–ensemble paradigm is shown in Figure 1. The three main steps of the ensemble model are as follows:

-Step 1: Decomposition process:

First, the features and noise of the original pollution data needed to be cleaned and processed so that an effective prediction model could be built. In this study, VMD technology was used to disaggregate the original pollution datasets into a set of VMs and the residue component with corresponding frequencies.

-Step 2: Ensemble forecasting and IWOA optimization:

The decomposition sequences with different characteristics were obtained via the VMD process. However, different sequences had different properties, which meant that a single prediction method could no longer effectively adapt to all the characteristics of the VMs. Thus, the ensemble strategy is adopted to solve this problem, and can be described as that if there are *M* types of prediction methods with the correct selection of weight coefficients to solve a problem. The results of multiple models were added together. Assume that $E_{model}$ (Model = “BPNN”, “ANFIS”, “ANFIS-FCM”, “GMDH”) is the ensemble prediction result of each VM by using the above methods. Then, using IWOA to optimize the output of the $E_{model}$, it can be expressed as
(18)$$Output_{SCWOA-NNCT}^{VMs}=w_{1}\times P_{model1}+w_{2}\times P_{model2}+w_{3}\times P_{model3}\ldots$$
where $w_{i}$ (*i* = 1, 2, …, *N*) is the weight coefficient of the model *N*. $w_{i}\in \left[-2,\;2\right]$ is the range of weight coefficients by NNCT [27].

To improve the optimal weight coefficients $w_{i}$(*i* = 1, 2, …, *N*), IWOA was employed to find the optimal solution for the ensemble weight coefficients. Before optimization, the objective equation needed to be confirmed first. The objective function of this paper is set by Equation (18). When the predefined minimum value of the objective function or the maximum iterations was reached, the optimization process was terminated. Nevertheless, the search boundary of the WOA is set to [−2, 2], the nesting dimension is 5 and the maximum number of iterations is 500.

-Step 3: Assemble forecasting results:

Through the above steps, the overall prediction results of the VMs were obtained. Then, the prediction results were combined to obtain the final result.

## 4. Study Areas and the Evaluation Criteria

### 4.1. Data Description

In this paper, the PM_2.5_ concentration data from the Environmental Protection of the People’s Republic of China (http://www.mep.gov.cn/) were collected to verify the performance of the proposed model. The selected daily PM_2.5_ concentration sample data are for Beijing, Tianjin, Baoding and Shijiazhuang from 1 August 2015 to 31 August 2017. The total data number of daily PM_2.5_ concentration for each city were 763. In each experiment, the first 572 data (approximately 75% of the total data) of each VM were used for training subsection, and the rest were the test subsection. When all predicted VMs were integrated into the overall result, the 189 pieces of data (about 25%) in the test results were used for optimizing the weights of the ensemble model and the rest were used for model testing.

### 4.2. Model Assessment Standards

To effectively assess the prediction performance of the developed model, four popular error criteria, shown in Table 1, were employed to assessment the prediction capacity of the developed model. Smaller values denote better prediction performance.

Here, $y_{n}$ and ${\hat{y}}_{n}$ present the actual and predicted values at time *n*, respectively. *N* denotes the sample size.

## 5. Results and Analysis

### 5.1. Data Decomposition by VMD

In the proposed VMD-IWOA ensemble model, the original PM_2.5_ concentration sequence is first decomposed into several independent VMs by using VMD. However, too many VMs introduce new problems. During the integrated prediction process, each VM generates estimation errors, and too many VMs cause an accumulation of errors. It also increases the time consumed in a single prediction step. To prevent the above problems, the entire VMs were restructured into three VMs and a residual.

### 5.2. The Process of Ensemble Forecast on VMs

The BPNN, ANFIS, ANFIS-FCM and GMDH prediction models were applied to forecast each VM, which reconstructed in Section 5.1. Additionally, then, the ensemble model integrates the results of the four prediction models on each VM, and optimizes the weights of the four prediction results based on IWOA. Before the simulation, the parameters of the four neural network model need to be initialized. The input nodes of the neural network are set to four, the hidden nodes to nine and the output nodes to 1. Besides, the rolling single-step forecasting operation method based on PM_2.5_ concentration data of four cities is used to test the predictive performance. The detailed experimental parameters of the four neural networks are shown in Table 2.

Table 3 shows the prediction results of the single models and the proposed ensemble model for each VM. To evaluate model performance, the RMSE was utilized as a model evaluation index. As can be seen from Table 3, each model performed optimally predictive behavior at a particular VM. For instance, the experimental results in Beijing were shown as follows: the BPNN provides the lowest RMSE values among all single models at VM2 and VM3, while at VM1 and residual, GMDH has the lowest RMSE values. The prediction results in Tianjin show that the ANFIS presents the best results at VM1. The FCM performs best at VM3. At VM2 and Residual, the GMDH provides the best results. The experimental results in Baoding show that among all of the single models, the RMSE value was lower than those of the other methods at VM1 and Residual, when the ANFIS was applied. At VM2 and VM3, the GMDH presents the optimal results. The forecasting results in Shijiazhuang reveal that the GMDH performs better than the others at VM2, VM3 and residual while ANFIS performs the best at VM1.

Based on the above analysis, it can be revealed that each model has its advantages on the particular VMs. A single prediction model cannot be used to predict all decomposition signals uniformly. Thus, the most suitable model is selected according to the different conditions, which reveals that an ensemble model can incorporate the virtues of multiple individual models to overcome the limitations of individual models. Therefore, this study proposed an ensemble model based on the IWOA to seek the best weight coefficients of the ensemble model. The searching boundary is set in [−2, 2] based on the NNCT, and the RMSE criteria is used as fitness function of IWOA. Table 3 presents the best weights and final results of the ensemble model. By comparing with each single model, it indicates that the developed ensemble model can give the desired prediction results.

Comparing the ensemble model with BPNN, ANFIS, FCM and GMDH, the average RMSE of four cities at VM1 was reduced by 26.10%, 2.62%, 7.80% and 3.97%, respectively; At VM2, the average RMSE of four cities was reduced by 5.81%, 11.15%, 11.51% and 3.59%; At VM3, the average RMSE of four sites was reduced by 7.19%, 58.21%, 13.92% and 6.22%, respectively. For Residual, the average RMSE of four sites was reduced by 17.79%, 33.09%, 22.27% and 7.51%, respectively. Consequently, it can be seen that compared with the single models BPNN, ANFIS, FCM and GMDH, the forecasting result of the ensemble model is significantly improved on each VM component.

### 5.3. Model Performance Evaluation and Comparison

To evaluate the proposed ensemble model, three types of model comparison experiments were designed to compare the proposed ensemble model with other individual models, VMD-based models, and existing benchmark models.

#### 5.3.1. Experiment 1: The Comparison between the Ensemble Model and VMD-Based Models

The experiment compares four VMD-based prediction models with the developed ensemble model. The four VMD-based models are VMD-BPNN, VMD-ANFIS, VMD-FCM and VMD-GMDH, which were constructed to emphasize important usages of the data decomposition technology. The corresponding improvement of the developed ensemble model and the VMD-based models are shown in Table 4 and Figure 2. By comparing the ensemble model with the VMD-BPNN, VMD-ANFIS, VMD-FCM and VMD-GMDH, we can conclude that the ensemble model significantly outperforms the other VMD-based models according to four evaluation criteria. For example, in Beijing, the ensemble model leads to 2.3843, 10.6660, 3.6867 and 2.1953 reductions in MAE, 5.4454, 21.3895, 11.6926 and 9.7510 reductions in RMSE, 0.3159, 11.9748, 12.3061 and 5.3553 reductions in MAPE, 5.2795, 21.3508, 11.6465 and 9.6318 reductions in TIC to compare with VMD-BPNN, VMD-ANFIS, VMD-FCM and VMD-GMDH, respectively. In addition, Figure 2 illustrates the comparison of actual values and the forecast values. The predicted results from the developed ensemble model are better than other VMD-based models.

#### 5.3.2. Experiment 2: The Comparison between the Ensemble Model and Individual Models

This experiment used four individual models to make comparison with the developed ensemble model. The four individual models are BPNN, ANFIS, FCM and GMDH. Table 5 indicates the comparison forecasting results between ensemble model and other single models. From Table 5, by comparing the ensemble model with the BPNN, ANFIS, FCM and GMDH, there are significant improvements in the predictions of the proposed model. For example, in Beijing, the ensemble model leads to 66.4829, 71.3965, 67.8848 and 82.5946 reductions in MAE, 65.7865, 73.3943, 7.7401and 81.1458 reductions in RMSE, 67.7547, 71.7270, 67.5358 and 83.5715 reductions in MAPE, 65.6355, 73.1804, 67.5553 and 80.7598 reductions in TIC to compare with BPNN, ANFIS, FCM and GMDH, respectively. Besides, Figure 3 presents the comparison between the actual values and the forecast values. The forecast results from the developed ensemble model are better than other single models.

#### 5.3.3. Experiment 3: The Comparison between the Proposed Model and the Existing Models

This part was conducted to further verify that the suggested hybrid decomposition–ensemble method can effectively improve performance prediction. Several existing models widely used in environmental prediction were applied to conduct comparative studies to access the suggested models. The existing models include two simple algorithms (i.e., ARIMA and RBFNN) and three hybrid algorithms (i.e., SSA-ENN, EEMD-GRNN and EEND-WOA-BPNN). The results of the comparative study are given in Table 6 and Figure 4. It can be seen from Table 6 and Figure 4 that the values of MAE, RMSE, MAPE and TIC of the developed model are all lower than the other existing models, which further shows the prediction performance of the developed ensemble model has obvious advantages. For example, comparing the proposed model with ARIMA, RBFNN, SSA-ENN, EEMD-GRNN and EEND-WOA-BPNN, the MAPE of Beijing was reduced by 94.01%, 91.14%, 90.22%, 86.73% and 69.20%, respectively. For Tianjing, the average MAPE of was reduced by 92.13%, 89.47%, 89.39%, 82.92% and 57.96%, respectively. For Baoding, the average MAPE was reduced by 94.01%, 91.14%, 90.22%, 86.73% and 69.20%, respectively. For Shijiazhuang, the average MAPE was reduced by 94.01%, 91.14%,90.22%, 86.73% and 69.20%, respectively.

In addition, the error mean and error STD are also used to evaluate the models’ accuracy and stability, and the results shows that the developed model has higher accuracy and stability than other existing models. Therefore, it can be concluded that the proposed ensemble model can be successfully and effectively employed for PM_2.5_ concentration prediction compared with existing models. Furthermore, the proposed ensemble model has the following highlights compared to previous works [15,16]: 1. the data decomposition; 2. multi-model integration prediction; and 3. the optimized ensemble pattern weighting coefficients.

## 6. Conclusions

Reliable and precise PM_2.5_ concentration forecasting is important for air quality early warning and pollution control. Owing to uncertainties and unstable of the PM_2.5_ datasets, the original PM_2.5_ series are very difficult to forecast accurately. Thus, it is still a challenging task to predict and simulate the PM_2.5_ reasonably. In this study, a new hybrid decomposition–ensemble learning paradigm, which based on variation mode decomposition (VMD) and modified whale-optimization algorithm (IWOA), is proposed to predict the PM_2.5_ concentration. In this developed paradigm, the VMD method was employed to decompose the original PM_2.5_ sequence into several VM series for forecasting. The prediction results show that the single prediction model used for pollution concentration prediction has limited capability and is not appropriate for all VMs. To this end, an ensemble model, based on four individual forecasting approaches, BPNN, ANFIS, FCM and GMDH, is proposed for predict all the VM components. Furthermore, in order to ascertain the best ensemble weight coefficients, an improved Whale Optimization Algorithm, named IWOA, is proposed and the final forecasting results were achieved by reconstructing the precise sequence. The main contributions of this paper are summarized as follows: (1) A new decomposition–ensemble learning paradigm is developed for PM_2.5_ concentration forecasting. (2) The VMD technique is adopted to decompose the primary PM_2.5_ series. (3) ANFIS, ANFIS-FCM and GMDH are utilized for PM_2.5_ forecasting. (4) An improved heuristic algorithm, IWOA, is developed to improve the weight coefficients of the ensemble model.

To evaluate the developed model, daily PM_2.5_ sequence from four cities located in Jing-Jin-Ji area of China were collected as the test cases for the comparison study. The comparison results indicated that the developed ensemble model is superior to comparison models, include four VMD-based models, four individual models, two benchmark models and three existing models. Thus, the developed ensemble model provides an effective forecasting ability, especially for the highly volatile and irregular data (e.g., PM_2.5_ concentration) and can be a powerful tool for decision makers in air quality monitoring and early warning system.

## Figures and Tables

**Figure 1 ijerph-18-01024-f001:**
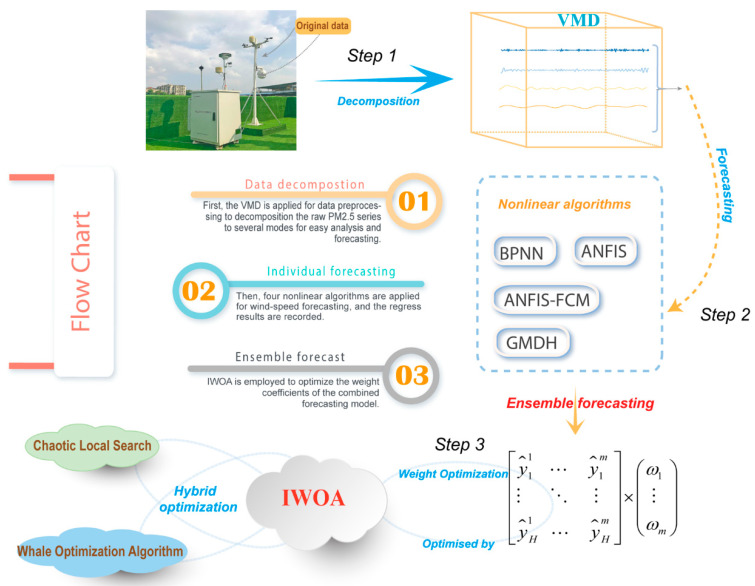
The main procedures of the combinatorial model proposed in the paper. Step 1: Variation mode decomposition (VMD) process; Step 2: Ensemble forecasting on variational modes (VMs); Step 3: Assemble forecasting results.

**Figure 2 ijerph-18-01024-f002:**
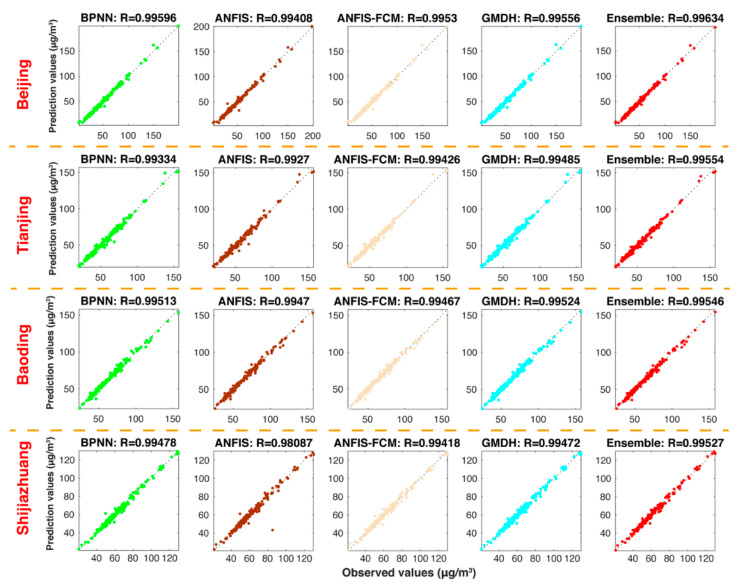
Forecasting results of the ensemble model and VMD-based models.

**Figure 3 ijerph-18-01024-f003:**
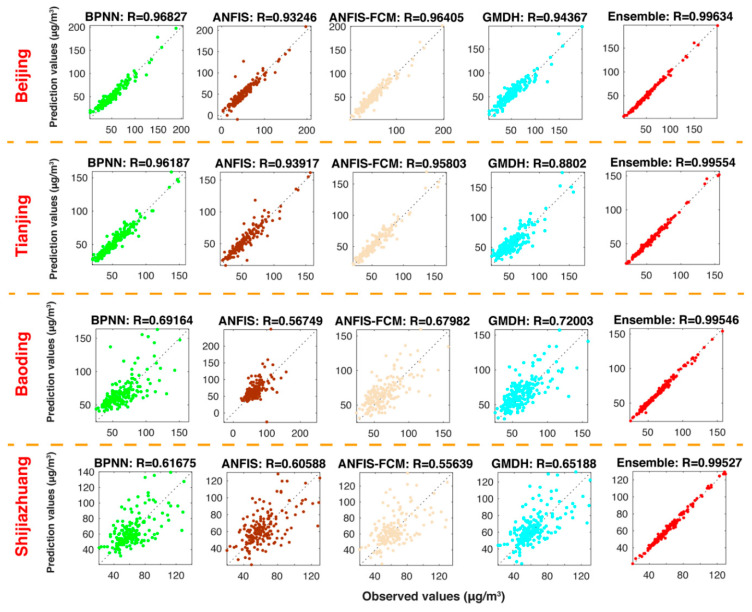
Forecasting results of the ensemble model and individual models.

**Figure 4 ijerph-18-01024-f004:**
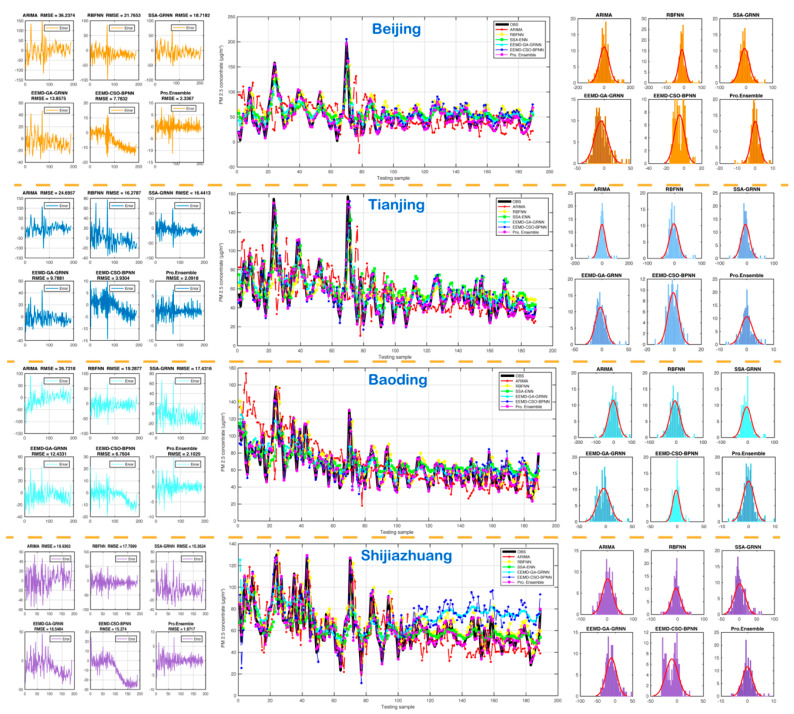
Forecasting results of the ensemble model and existing models.

**Table 1 ijerph-18-01024-t001:** Four evaluation rules.

Metric	Equation	Definition
**MAE**	$\mathrm{MAE}=\frac{1}{N}\sum_{n=1}^{N}\left|y_{n}-{\overset{\wedge }{y}}_{n}\right|$	The average absolute forecast error of *n* times forecast results
**RMSE**	$\mathrm{RMSE}={\left(\frac{1}{N}{\sum_{n=1}^{N}\left(y_{n}-{\overset{\wedge }{y}}_{n}\right)}^{2}\right)}^{1/2}$	The root mean-square forecast error
**MAPE**	$\mathrm{MAPE}=\frac{1}{N}\sum_{n=1}^{N}\left|\frac{y_{n}-\overset{\wedge }{y_{n}}}{y_{n}}\right|\times 100\%$	The average of absolute error
**TIC**	$\mathrm{TIC}=\frac{\sqrt{\frac{1}{N}{\sum_{n=1}^{N}\left(y_{n}-{\hat{y}}_{n}\right)}^{2}}}{\sqrt{\frac{1}{N}\sum_{n=1}^{N}y_{n}{}^{2}}+\sqrt{\frac{1}{N}\sum_{n=1}^{N}{\hat{y}}_{n}{}^{2}}}$	Theil’s inequality coefficient

**Table 2 ijerph-18-01024-t002:** Experiment parameters of artificial neural networks (ANNs).

Model	Experimental Parameters	Default Value
BPNN	The learning velocity	0.01
	The maximum number of trainings	1000
	Training requirements precision	0.00004
ANFIS	Spread of radial basis functions	0.5
	Training requirements precision	0.00004
FCM	The maximum number of trainings	1000
	Spread of radial basis functions	0.15
GMDH	Learning rate	0.1
	Training requirements precision	0.00004

**Table 3 ijerph-18-01024-t003:** VMs forecasting results of the individual model and ensemble models in four cities.

	Models	VM1		VM2		VM3		Residual	
		Weights	RMSE	Weights	RMSE	Weights	RMSE	Weights	RMSE
**Beijing**	BPNN	0.06880	0.32695	0.40850	0.67043	1.24076	0.81130	0.51770	1.16300
	ANFIS	0.30331	0.30256	0.03701	0.75516	−0.10014	1.38320	0.38167	1.17560
	ANFIS-FCM	0.00617	0.30669	0.17178	0.73279	0.18866	0.87710	−0.48722	1.43640
	GMDH	0.63306	0.29672	0.38893	0.68048	−0.34365	0.86398	0.56177	1.16110
	Ensemble model	-	0.29096	-	0.65445	-	0.79141	-	1.05820
**Tianjing**	BPNN	0.03703	0.31010	0.11645	0.61997	0.16865	0.79890	−0.57673	1.34090
	ANFIS	0.34467	0.25575	0.35167	0.64571	−0.17093	0.90740	−0.00375	1.73350
	ANFIS-FCM	0.26816	0.25998	−0.07272	0.66214	0.74391	0.75810	−0.14957	1.23630
	GMDH	0.34997	0.25685	0.61707	0.59548	0.24071	0.78005	1.71433	0.99831
	Ensemble model	-	0.24593	-	0.57482	-	0.73581	-	0.93073
**Baoding**	BPNN	−0.04298	0.36294	−0.35375	0.65537	0.11825	1.19390	0.17385	0.82444
	ANFIS	0.72431	0.26715	0.33840	0.67208	0.12688	1.49690	0.40790	0.80775
	ANFIS-FCM	0.10233	0.28678	0.33900	0.65961	−0.03061	1.35590	−0.00282	0.85015
	GMDH	0.21607	0.27619	0.67852	0.65221	0.81967	1.13750	0.38544	0.86024
	Ensemble model	-	0.26360	-	0.63175	-	1.08830	-	0.77429
**Shijiazhuang**	BPNN	−0.07031	0.29393	−1.28208	0.60497	0.40346	1.09080	−0.01858	0.78471
	ANFIS	1.01576	0.23183	0.50000	0.61216	−0.04074	2.99250	−0.07890	0.92644
	ANFIS-FCM	−0.08646	0.25504	−0.20145	0.63428	0.15397	1.18430	0.32321	0.78460
	GMDH	0.14144	0.23988	2.00000	0.57380	0.50412	1.06280	0.75931	0.72264
	Ensemble model	-	0.22918	-	0.55405	-	1.01030	-	0.70881

**Table 4 ijerph-18-01024-t004:** The results of the ensemble model and other VMD-based models at four cities.

Dataset	Indicator	Ensemble Model vs. VMD-BPNN	Ensemble Model vs. VMD-ANFIS	Ensemble Model vs. VMD-ANFIS-FCM	Ensemble Model vs. VMD-GMDH
**Beijing**	IMAE (%)	2.3843	10.6660	3.6867	2.1953
	IRMSE (%)	5.4454	21.3895	11.6926	9.7510
	IMAPE (%)	0.3159	11.9748	12.3061	5.3553
	ITIC (%)	5.2795	21.3508	11.6465	9.6318
**Tianjing**	IMAE (%)	14.0270	14.5473	7.3431	3.5937
	IRMSE (%)	18.0484	21.9939	11.8452	7.1982
	IMAPE (%)	14.5592	12.9259	10.8481	3.3801
	ITIC (%)	18.0279	22.0097	11.7854	7.1898
**Baoding**	IMAE (%)	2.8295	7.3044	6.9863	0.5752
	IRMSE (%)	3.9004	7.4784	7.6538	2.3325
	IMAPE (%)	2.1528	7.0565	6.0185	0.0488
	ITIC (%)	3.8167	7.5020	7.6881	2.3468
**Shijiazhuang**	IMAE (%)	1.4251	18.7068	4.7226	4.8292
	IRMSE (%)	4.8678	49.9864	9.2788	5.0535
	IMAPE (%)	1.2716	15.5030	5.3138	4.0961
	ITIC (%)	4.8218	50.0715	9.3976	5.0563

**Table 5 ijerph-18-01024-t005:** The results of the ensemble model and other single models at four cities.

Dataset	Indicator	Ensemble Model vs. BPNN	Ensemble Model vs. ANFIS	Ensemble Model vs. ANFIS-FCM	Ensemble Model vs. GMDH
**Beijing**	IMAE (%)	68.6537	71.9248	69.1895	80.9890
	IRMSE (%)	66.6395	76.9644	68.1596	78.5772
	IMAPE (%)	69.6792	68.0098	58.9320	79.7105
	ITIC (%)	66.3829	76.7079	67.9496	77.7231
**Tianjing**	IMAE (%)	66.4829	71.3965	67.8848	82.5946
	IRMSE (%)	65.7865	73.3943	67.7401	81.1458
	IMAPE (%)	67.7547	71.7270	67.5358	83.5715
	ITIC (%)	65.6355	73.1804	67.5553	80.7598
**Baoding**	IMAE (%)	87.9473	90.1459	89.0888	88.2149
	IRMSE (%)	88.0371	90.9558	88.0382	87.3972
	IMAPE (%)	88.0240	89.6555	89.2647	88.3508
	ITIC (%)	87.7416	90.6394	87.7908	87.0859
**Shijiazhuang**	IMAE (%)	88.3384	88.7396	89.1616	88.0327
	IRMSE (%)	88.8181	88.9320	89.5788	88.2980
	IMAPE (%)	87.7574	88.4450	88.8123	87.8461
	ITIC (%)	88.8018	88.8921	89.4880	88.1406

**Table 6 ijerph-18-01024-t006:** Comparison of prediction performances with existing models.

Dataset	Indicator	MAE	RMSE	MAPE	TIC	Error Mean	Error STD
**Beijing**	ARIMA	26.4865	36.2374	92.8296	0.3176	−0.1317	36.3334
	RBFNN	17.0813	21.7653	62.7264	0.1751	−11.9703	18.2263
	SSA-ENN	14.8237	18.7182	56.8534	0.1593	−6.4266	17.6271
	EEMD-GRNN	11.3569	13.8575	41.9101	0.1176	−6.0212	12.5142
	EEMD-WOA-BPNN	6.3748	7.7832	18.0544	0.0647	−5.2481	5.7629
	Pro. Ensemble	1.6843	2.3367	5.5600	0.0202	0.0577	2.3422
**Tianjing**	ARIMA	17.4613	24.6957	35.4034	0.2059	−1.6273	24.7075
	RBFNN	12.5062	16.2787	26.4680	0.1376	−2.2933	16.1591
	SSA-ENN	12.4636	16.4413	26.2710	0.1335	−5.6593	15.4776
	EEMD-GRNN	7.6808	9.7881	16.3186	0.0815	−2.8509	9.3886
	EEMD-WOA-BPNN	3.2047	3.9304	6.6289	0.0328	−0.8012	3.8581
	Pro. Ensemble	1.4745	2.0918	2.7869	0.0176	0.0061	2.0973
**Baoding**	ARIMA	20.0634	26.7218	31.9956	0.1888	−3.9880	26.4927
	RBFNN	15.0370	19.2877	25.5808	0.1376	−5.4023	18.5648
	SSA-ENN	13.4114	17.4316	24.1157	0.1275	−4.2010	16.9627
	EEMD-GRNN	9.5818	12.4331	17.2851	0.0902	−4.2821	11.7035
	EEMD-WOA-BPNN	5.0988	6.7604	9.2280	0.0495	−2.2183	6.4030
	Pro. Ensemble	1.4926	2.1029	2.4427	0.0156	−0.0498	2.1079
**Shijiazhuang**	ARIMA	15.5991	19.9363	25.4303	0.1541	1.9553	19.8929
	RBFNN	13.4980	17.7899	23.6275	0.1310	−5.9717	16.8022
	SSA-ENN	11.1319	15.3624	18.5469	0.1204	0.9266	15.3752
	EEMD-GRNN	15.5582	18.5484	29.2907	0.1326	−11.1552	14.8584
	EEMD-WOA-BPNN	11.9964	15.2740	22.6224	0.1093	−9.8333	11.7187
	Pro. Ensemble	1.4004	1.8717	2.3930	0.0143	−0.0449	1.8762

## Data Availability

In this paper, the PM_2.5_ concentration data from the Environmental Protection of the People’s Republic of China (http://www.mep.gov.cn/) were collected to verify the performance of the proposed model.

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
