# Peer review of "Research on a Novel Hybrid Decomposition–Ensemble Learning Paradigm Based on VMD and IWOA for PM2.5 Forecasting"

_ijerph, 2021, doi:10.3390/ijerph18031024_

Round 1

Reviewer 1 Report

I think that this manuscript can be accepted for published in this journal for its current revised form.

Reviewer 2 Report

Accept

This manuscript is a resubmission of an earlier submission. The following is a list of the peer review reports and author responses from that submission.

Round 1

Reviewer 1 Report

This research proposed a novel hybrid decomposition-ensemble learning paradigm based on variation mode decomposition and improved whale-optimization algorithm  to address complex nonlinear environmental data, its very useful for improving prediction accuracy. This paper provides new method to prediction the PM2.5.  The structure of the paper is very reasonable. But, in my opinion, this paper is need major revision before publication.

Issues:

Line 89-90 merge together or delet The WOA and IWOA are described in Section 2.3.

Line 95-101 delete

Line 228 fig 2? Where is fig 1?

Line 336 fig 4 where is fig 3?

Reviewer 2 Report

This paper present a hybrid model using hybrid decomposition-ensemble learning paradigm based on VMD and IWOA for PM2.5 forecasting. The proposed method is verifed on 4 realistic datasets and compared with other methods. The obtained results evidenced the feasibility of the proposed model. I suggest this paper for publication after addressing the following comments.

1. Very high similarity index. This is the most critical point of this paper. The reviewer uses turnitin to check the similarity of this paper. Results are 50% if including the reference list and 41% if excluding the list.

2. The literature review is lack of one important paper of the authors on the topic. “Day-Ahead PM2.5 Concentration Forecasting Using WT-VMD Based Decomposition Method and Back Propagation Neural Network Improved by Differential Evolution”, by Wang et al. 2017. This is a hybrid model. To show its performance, the proposed method should be compared with the Wang’s model and other similar hybrid models. Which one is better?

3. This paper is about proposing a hybrid model for PM2.5 prediction. Thus, other hybid models should be reviewed critically.

Reviewer 3 Report

  1. The authors should add the related references in the years of 2019 ~ 2020 in the introduction section of the revised manuscript.
  2. This manuscript should also ask for a native English professor to help edit the whole text. It should include the whole manuscript and the grammar.
  3. What’s the perspective industrial applications for this study?
  4. This study can be accepted for published in this journal after the above minor comments have been addressed.

Reviewer 4 Report

This paper is a useful addition to the existing knowledge literature. Authors have done interesting work presenting the advanced methods to forecast the PM2.5. The novel hybrid approaches proposed by the paper provides a significant contribution to the domain. Furthermore, the authors' effort to bring forward the ensemble approach to overcome single methods limitation is also an interesting contribution. Overall, the contribution is well articulated, and the methodology is well presented. However, there are a few aspects that the author might need to consider to improve the present contribution. Below are my few comments which the author need to consider:

  1. Introduction: Minor grammatical corrections are expected, for example, line 28,47,60,77,93,101. A similar revision needs to be made on various occasions. The full form of the abbreviations is missing when first used (Ex: line 60).
  2. Related Methodology: Equation 3 "u" was not introduced. Abbreviations were not introduced on the occasions where they were first used ( line 107). Grammar corrections are needed (for example, line 233, 247, etc. ). The flow chart of methodology can be well presented with a UML diagram to make it more understandable the flow. 
  3. Result and Analysis: Like the previous section, the correction is needed in terms of grammar and references to Figure (where is figure 1(a)? line 284). I suggest the author make the comparison more visual than just tables so that comparison can be easily grasped. Also, the images need to be resized to make them readable ( Ex: Figure 4).
  4. I suggest the authors present the discussion section highlighting the limitations and comparison of the results obtained with other existing studies to make the novelty very clear. Particularly, the authors need to consider presenting the limitation in terms of the ensemble, as every single method in the ensemble carries some form of bias, and how it was overcome is not well explained. 

Overall, minor work is needed to make the paper more compact.